# From Real-World Data to Causally Interpretable Models: A Bayesian Network to Predict Cardiovascular Diseases in Adolescents and Young Adults with Breast Cancer

**DOI:** 10.3390/cancers16213643

**Published:** 2024-10-29

**Authors:** Alice Bernasconi, Alessio Zanga, Peter J. F. Lucas, Marco Scutari, Serena Di Cosimo, Maria Carmen De Santis, Eliana La Rocca, Paolo Baili, Ilaria Cavallo, Paolo Verderio, Chiara M. Ciniselli, Sara Pizzamiglio, Adriana Blanda, Paola Perego, Paola Vallerio, Fabio Stella, Annalisa Trama

**Affiliations:** 1Evaluative Epidemiology Unit, Department of Epidemiology and Data Science, Fondazione IRCCS Istituto Nazionale dei Tumori, 20133 Milano, Italy; annalisa.trama@istitutotumori.mi.it; 2Department of Informatics, Systems and Communication, University of Milano-Bicocca, 20100 Milano, Italy; a.zanga3@campus.unimib.it (A.Z.); fabio.stella@unimib.it (F.S.); 3Data Science and Advanced Analytics, F. Hoffmann—La Roche Ltd., 4070 Basel, Switzerland; 4Datascience Department, University of Twente, 7500 Enschede, The Netherlands; p.j.f.lucas@utwente.nl; 5Istituto Dalle Molle di Studi Sull’intelligenza Artificiale (IDSIA), 6962 Lugano, Switzerland; scutari@bnlearn.com; 6Department of Advanced Diagnostics, Fondazione IRCCS Istituto Nazionale dei Tumori, 20133 Milano, Italy; serena.dicosimo@istitutotumori.mi.it; 7Radiation Oncology 1 Unit, Fondazione IRCCS Istituto Nazionale dei Tumori, 20133 Milano, Italy; mariacarmen.desantis@istitutotumori.mi.it; 8Department of Radiation Oncology, Azienda Ospedaliero Universitaria Integrata, 37126 Verona, Italy; eliana.larocca@aovr.veneto.it; 9Data Science Unit, Department of Epidemiology and Data Science, Fondazione IRCCS Istituto Nazionale dei Tumori, 20133 Milano, Italy; paolo.baili@istitutotumori.mi.it; 10Scientific Directorate, Fondazione IRCCS Istituto Nazionale dei Tumori, 20133 Milano, Italy; ilaria.cavallo@istitutotumori.mi.it; 11Unit of Bioinformatics and Biostatistics, Department of Epidemiology and Data Science, Fondazione IRCCS Istituto Nazionale dei Tumori, 20133 Milano, Italy; paolo.verderio@istitutotumori.mi.it (P.V.); chiara.ciniselli@istitutotumori.mi.it (C.M.C.); sara.pizzamiglio@istitutotumori.mi.it (S.P.); adriana.blanda@istitutotumori.mi.it (A.B.); 12Molecular Pharmacology Unit, Department of Experimental Oncology, Fondazione IRCCS Istituto Nazionale dei Tumori, 20133 Milano, Italy; paola.perego@istitutotumori.mi.it; 13Cardiology Unit, Fondazione IRCCS Istituto Nazionale dei Tumori, 20133 Milano, Italy; paola.vallerio@istitutotumori.mi.it

**Keywords:** adolescents and young adults, survivorship, breast cancer, cardiotoxic treatments, risk prediction, personalized follow-up, artificial intelligence

## Abstract

Cardiovascular diseases are among the most frequent, although rare, long-term sequalae in adolescents and young adult survivors of breast cancer. However, no dedicated tool exists to help clinicians with planning personalized follow-up strategies for these patients. To make up for this lack, in this work, we developed a Bayesian network, an artificial intelligence model, to predict the 5-year risk for cardiovascular diseases in these patients, leveraging real-world data from two different cohorts. The model showed a very good ability to identify patients at risk and select those that should be prioritized because they are at higher risk, making it useful for guiding clinicians in everyday practice. Finally, the methodological approach proposed in this work is particularly interesting for all researchers who aim at developing causally interpretable tools, also dealing with real-world data and their biases.

## 1. Introduction

The increasing number of cancer survivors in recent decades has moved researchers’ attention from cancer treatment to its long-term outcomes. In this context, artificial intelligence (AI) has achieved remarkable results in identifying cancer patients at risk of developing long-term outcomes, based on different types of data (such as omics and imaging data) [1,2]. Such results can be exploited to minimize therapy-related morbidity by offering tailored treatments and, at the same time, ensure the most appropriate management for high risk. However, AI models are known to be computationally intensive and to require large amounts of data to perform optimally. Moreover, real-world data (RWD) used to develop and train these models are often scarce, especially if the outcome of interest is rare or biased. Additionally, healthcare data are typically heterogeneous given the granularity of the information collected and are stored in silos which do not communicate with each other. In this framework, data fusion is needed to provide accurate predictions by combining heterogeneous data with expert knowledge.

Bayesian networks (BNs) are probabilistic graphical models which use graphical representations to describe conditional probability dependence relationships [3]. These AI models are well suited to prognostication and can be applied even when events are rare and some information is missing [4]. Moreover, BNs are able to merge information retrieved from different sets of data, handle the biases in RWD, and integrate domain knowledge coming from experts or the literature; all these properties make them an excellent candidate to build effective predictive models using RWD [5]. Moreover, besides the fact that the decision-making processes behind the most used AI tools are so complex as to make them perceived as black boxes by humans, BNs allow for the study of the cause–effect relationships of all the variables with one another and thereby resemble clinical reasoning [6].

Adolescents and young adults (AYAs) (15–39 years at first cancer diagnosis) are a heterogeneous group of cancer patients whose needs, expectations, and treatment deserve special consideration [7]. As in older women, breast cancer (BC) is the most common cancer in AYA females, with very high survival rates [8]. Compared to their older counterparts, however, female AYA patients present more aggressive forms of BC [9,10]. Cardiovascular diseases (CVDs) are the most common, although rare, long-term outcome in these patients. The main causes of CVDs in these survivors (as in older women) include not only intensive cardiotoxic treatments [11,12,13] but also lifestyle and socioeconomic factors (e.g., smoking, obesity) and genetics [14,15].

Despite the large availability of data on older BC patients, no study has characterized the CVD risk in AYAs. Nevertheless, predictive AI tools developed for older patients are not directly transportable to this peculiar group of younger patients. Against this background, the project “*pRedicting cardiOvascular diSeAses iN adolescent and young breast caNcer pAtients*” (ROSANNA) is aimed at building a causally interpretable AI-based model to identify AYA BC survivors at higher risk of developing CVDs. In practice, we used a BN to estimate the CVD risk by solving a maximum a posteriori query via approximate inference. This allowed us to leverage the patient-specific conditions given by the observed covariates, mitigating the impact of the missing ones.

## 2. Materials and Methods

### 2.1. Population-Based Cohort (PBC)

A retrospective cohort of about 70,000 AYA patients was collected in the context of a previous project, named the Ada cohort [16]. To develop this cohort, 34 Italian population-based cancer registries (CRs) were asked to identify AYA cases and link several administrative datasets (i.e., hospital discharge records, outpatients, and drug flows) for long-term outcome identification. Among the Ada cohort, we identified 1036 BC patients, diagnosed between 2009 and 2014, with the following criteria:Female patients aged 18–39 years at first BC diagnosis;One primary tumor only;Malignant carcinomas only;1-year survivors;No metastases at diagnosis;No CVD before cancer diagnosis.

We retrieved information on treatments, CVDs, and major cardiovascular risk factors from the administrative databases based on previously used, validated algorithms [17,18], described in the Appendix A.

### 2.2. Clinic-Based Cohort (CBC)

The PBC had several advantages, such as the high number of patients and the possibility of tracking patients for long-term outcomes, but it lacked cancer prognostic factors and treatment details that are essential to drive causal interpretations. To overcome this limitation, we collected an additional retrospective single-institution (Fondazione IRCCS Istituto Nazionale dei Tumori, Milano, Italy) CBC of 339 AYA BC patients, diagnosed in 2011–2019. Patients were identified in the INT BC registry applying the same criteria as in the population-based cohort, and detailed clinical information on the BC treatment and BC prognostic factors was collected via the clinical charts. In contrast to the PBC, in the CBC, it was not possible to link the administrative datasets; thus, the information on the CVD follow-up is completely missing. Data from the two cohorts were not directly linkable because, even though some patients may overlap, privacy constraints made it impossible to de-identify the patients.

### 2.3. Variables Analyzed

As opposed to standard statistical techniques, the properties of the BN model make it possible to deal with data sparsity and low numbers of events; thus, no formal statistical restriction was made on the list of variables to be included in the model. The selection of the variables was made according to expert opinion to maximize the model’s causal interpretability. All the variables selected for the analysis are described in Table 1.

Preoperative clinical and histopathological variables with prognostic value for the prediction of the 5-year survival of the tumor were identified according to standard treatment guidelines [19].

BC treatments (chemotherapy, radiotherapy, target therapy, and hormone therapy) were divided, according to the date of surgery, between neoadjuvant (i.e., pre-surgery) and adjuvant (i.e., post-surgery) treatments. All the treatments can, through different mechanisms, induce several types of cardiac events, which were further stratified as cardiac toxicities (i.e., conduction disorders, arrhythmias, or heart failure) and ischemic heart diseases (i.e., acute myocardial infarction or other forms of ischemic heart disease). In more detail, chemotherapy, radiotherapy, and targeted drugs exert a wide range of effects on the cardiac structure and function, with damage to cardiomyocytes and/or endothelial cells, resulting in various clinical alterations, including left-ventricular dysfunction, coronary artery diseases, pericarditis and pericardial effusion, cardiomyopathy, valvular disease and arrhythmias, and, in turn, heart failure in the most severe cases.

Hormone therapy blocks the body’s ability to produce hormones, particularly estrogen, which has a cardioprotective effect in young females. Thus, the loss of estrogen during menopause is associated with an increased risk of ischemic heart disease either directly through vascular cells and cardiac myocytes or indirectly through systemic pathways, such as by altering lipid profiles and inducing major cardiovascular risk factors such as hypertension and dyslipidemia.

The distribution of the two cohorts’ characteristics is illustrated in the Appendix A. Missing data significantly impacted this analysis and depended strongly on the cohort of origin (missing not at random); thus, for example, cancer prognostic factors are completely missing in the PBC, and the same applies to the CVD follow-up in the CBC.

### 2.4. Model Development and Evaluation

The workflow for the model development is illustrated in the Appendix A. A BN consists of two parts: (i) a directed acyclic graph (DAG) and (ii) a set of conditional probability tables (CPTs) [20]. The DAG represents dependencies among the variables, and the conditional probability distributions quantify these dependencies. Our work focused on a discrete BN, such that all the included variables were dichotomous or discrete (as shown in Table 1). Given the limited follow-up period of the most recently diagnosed patients, we set a 5-year time span for the CVD follow-up. We split the overall combined dataset (PBC + CBC) into two subsets: 339 patients from the CBC + 623 patients from the PBC (approx. 70% of the overall sample) contributed to the training set, while the remaining 413 patients from the PBC (approx. 30% of the overall sample) contributed to the validation set. The CBC did not include CVD information, so it contributed to the training set only.

Learning the BN model only from training data was not advisable due to the rarity of the events. To overcome data scarcity, we firstly designed a basic structure of the DAG using prior knowledge taken from the domain literature and discussed with clinical experts. Secondly, we used the Structural Expectation–Maximization algorithm to handle the missing-not-at-random issue. Structural Expectation–Maximization works when data are missing completely at random (MCAR) and missing at random (MAR) but can also be used when data are missing not at random (MNAR) if the missingness mechanism is modeled correctly, as in our case since the initial DAG was specified using prior knowledge. Then, the DAG was extended learning new causal relations directly from the training data (i.e., causal discovery) [21]. To ensure that the relationships identified in the RWD could be interpreted as causal, making it possible to leap from a probabilistic model to a causal BN, we subsequently validated them according to the existing literature before adding them as new arcs (i.e., arrows) to the model. We then estimated the CPTs using the training set. More technical details on the algorithms used to build the model are described in the work by Bernasconi et al. [22].

The validation set was used to validate the results and illustrate the possible model applications. To assess the model classification performance, i.e., its ability to correctly predict survivors not developing a CVD as not at risk and those developing a CVD as at risk, we used the standard classification metrics: the Area Under the Receiving Operator Characteristic Curve (AUC), True Positive Rate (TPR), True Negative Rate (TNR), Positive Predictive Value (PPV), Negative Predictive Value (NPV), Matthew’s correlation coefficient (MCC), and accuracy. Accuracy, although popular, can generate overoptimistic results, especially if the dataset suffers from class imbalance (e.g., rare events, such as CVDs in AYAs with BC). Accordingly, we also assessed the accuracy rebalanced for the minority class (i.e., balanced accuracy, or BA). References to all the metric formulas can be found in the paper by Chicco et al. [23]. We used a normal approximation for proportions to calculate 95% confidence intervals (CIs).

We plotted a lift chart to illustrate how the model predictions could be used to sort patients by their risk. In the validation set, we placed survivors in descending order of the predicted probability of developing a CVD (*x*-axis), while we plotted the corresponding observed proportion of survivors who actually developed a CVD, namely, the TPR, on the *y*-axis. The faster the growth of the cumulative gain curve, the better the model is at prioritizing those patients at higher risk. Finally, to support clinicians in everyday practice, we developed an app-based tool which allows for an estimation of the probability of the 5-year CVD risk for individual AYA BC patients.

For the BN development, training, and validation phases, we used the “bnlearn” package freely available in R [24].

## 3. Results

### 3.1. Model Structure and Validity

The DAG of the final model is shown in Figure 1. All the probability distributions are displayed using frequency bars in the nodes (i.e., squares), and the dependencies are indicated by the arcs connecting the nodes. Nodes linked to each other, either directly (when one direct arc connects them) or indirectly (when the arc connecting two nodes passes through a third node, namely, a mediator), are assumed to be dependent. If no direct or indirect arcs connecting two nodes are present, the nodes are assumed to be independent. Arcs pointing from one node toward another one indicate that the first node is the cause of the second (e.g., neoadjuvant treatments-->surgery). The direction of the arcs represents the flow of causation. The cohort of origin ([cohort])—PBC or CBC—is a context variable (i.e., it represents the context in which the data were collected). By placing this variable as the root node (i.e., the first node, with no incoming arc), we can describe the missing mechanism arising from the data’s different contexts of origin and allow the model to handle the selection bias; thus, patients in the CBC may differ from those in the PBC due to the selection mechanism. The node [death_in_5y], representing the patient’s death within 5 years of diagnosis, is included in the model to take into consideration its competing role with CVDs. In the upper part of the DAG, we included by design all the nodes representing the prognostic factors that mainly impact it; these nodes were included to describe which factors directly or indirectly determined the choice of neoadjuvant treatment and show their interplay, and how, in turn, they indirectly influenced the type of surgery and adjuvant treatment. The cardiotoxic role of chemotherapy, radiotherapy, and targeted drugs is illustrated in the left part of the DAG. On the opposite side, the hormone-induced risk of ischemic heart disease, hypertension, and dyslipidemia (which, together with type 2 diabetes, are among the major CVD risk factors) is described. The distribution of the latter may also depend on the cohort of origin, which accounts for the inclusion of the incoming arcs.

Figure 2 summarizes the overall performance of the BN. The BN model showed a high classification performance in the validation set, with an AUC, which is the measure of the model’s ability to distinguish between survivors at risk of developing a CVD and those not at risk, very close to 90%. The high level of accuracy (i.e., the probability of a given survivor being correctly identified as at risk or not at risk by the model) is strongly influenced by the perfect TNR (i.e., the probability of a survivor without a CVD being identified correctly by the model), being the negative value of the most frequent class of the target variable due to the rarity of the CVD event. In contrast, the model showed the lowest performance in terms of the TPR, which is the probability of a survivor with a CVD being identified correctly by the model, due to the low number of events the model was trained on. Thus, upon controlling for class imbalance and adjusting the accuracy accordingly (as it is for the balanced accuracy and MCC), the performances drop, even though they are maintained above 80%.

### 3.2. Clinical Applications

The lift chart (Figure 3) illustrates how the probability values given by the model can be used to sort BC survivors according to their risk of developing a CVD in the 5 years following cancer diagnosis. As illustrated in the figure, the cumulative gain curve (depicted in bold black) suddenly increases. Thus, focusing on the 25% of patients predicted by the BN model to be at higher risk of developing a CVD enabled the identification of approximately 81% of patients who will actually develop it. The latter percentage rises to 88% when focusing on the first 50% of patients.

Moreover, if the evidence is set, i.e., if the BN model is provided with information about an individual patient (for example, information about the neoadjuvant and/or adjuvant treatments received), the probability distributions are automatically updated, and the model allows for a prediction of the individual probability of developing a CVD within five years from the cancer diagnosis. To support clinicians in everyday practice, we designed and developed an app-based tool which allows for an estimation of the probability of 5-year CVD risk for individual AYA BC patients (the user interface of the application is shown in Figure 4).

## 4. Discussion

In this work, we developed and validated the first AI-based causal model for estimating the CVD risk in AYA females surviving BC. The model very accurately identified the patients who would develop CVDs (classification metrics), correctly ranking them by their risk probability (lift chart) and providing a causal interpretation of the variables involved. Moreover, using practical applications, we showed that the results obtained by our model could be pivotal for clinicians who aim at planning effective, tailored follow-up interventions for this particular group of patients.

### 4.1. Strengths

In the proposed BN, the context node accounted for the selection and missingness mechanism in the two cohorts, making it possible to combine them. Unfortunately, the same approach cannot be used for other machine learning (ML) techniques; therefore, standard ML models can only use multiple imputation approaches. Despite being very helpful, multiple imputation requires strong assumptions, such as data missing at random, which was not valid in this framework.

A crucial task for predictive models is to determine the minimal subset of variables needed to maximize the performance, which is what interests most clinicians and epidemiologists, especially when many variables are involved and the power set is limited. This can be easily performed in our model given its visual and interpretable causal structure, validated by medical knowledge [3].

Due to the AYA distinctive tumor case mix, biology, and treatments, it is not possible to identify, in the literature, any work that can be used for direct comparison. Despite this, the BN model proved very good at correctly prioritizing and classifying whether patients were at risk. In this setting, most classification metrics, such as the TNR and accuracy, are not informative, being biased by the rarity of CVDs in AYA BC survivors. The same also applies to the interpretation of the AUC, which is the classification metric most frequently presented in the scientific literature. Therefore, when dealing with rare events, preference should be given to metrics such as the balanced accuracy and MCC in view of their ability to summarize the classification performance while also considering the event prevalence [23].

Finally, classification is just one of the tasks that the BN model can accomplish. Indeed, we are currently working on applications of causal inference, such as the estimation of the causal effect of cancer treatments and counterfactual explanations [25].

### 4.2. Limitations

Like all AI models, the main limitations of this model are its assumptions.

First of all, the causal discovery algorithm used to train the BN has its own set of assumptions that must hold in order to obtain a sensible DAG. First, causal discovery assumes causal sufficiency. This assumption requires that all the causes related to the outcome of interest must be measured. In our case, being able to track adjuvant treatments, neoadjuvant treatments, type 2 diabetes, dyslipidemia, and hypertension allowed us to restrict the set of possible unobserved causes and investigate the potential impact of these variables on CVD. Nevertheless, this assumption is unlikely to hold in a real-world setting; hence, further methodological work is needed to extend the model to explicitly include latent confounders, like socioeconomic and lifestyle factors [26] and also genetic aspects [15,16] that can play an important role in the CVD risk of these patients. Moreover, it is not only important to include these quantities in the causal discovery process but also to correctly codify them so that the causal relationships are still valid. However, the expert domain knowledge required to perform this is difficult to achieve, especially in new, emerging fields such as cardio-oncology. Finally, causal discovery is known to be affected by the discretization process; hence, deciding the discretization thresholds is challenging and arbitrary.

Another strong assumption is the “no-multiple-versions-of-treatment assumption”, i.e., the definition of treatment is the same for all individuals. The BC treatments did not change during the considered period (diagnosis years 2009–2019); therefore, the treatments were not expected to be different in the CBC [19,27]. The same assumption may not be valid in the PBC, where different versions of treatment may be present, since the patients were treated in different hospitals. We do not expect major differences as national recommendations guide the treatment of BC patients and medical oncologists follow them not only for evidence-based standards but also for legal reasons. Nevertheless, attention should also be paid to the generalizability of the results to more recently diagnosed cases. Thus, the impact on the CVD risk profile of new schemes and therapies (e.g., immunotherapy) introduced in the 2020s needs to be further explored. In this regard, a new call for data collection on a more recent cohort of patients is envisioned in the next few years.

Finally, in advance of the validation performed internally in the data, the model needs external validation to confirm the transportability of the results to all AYAs with BC [28]. In this respect, work is already ongoing to externally validate the model in similar cohorts of AYA BC survivors in six different areas of Italy (Veneto, Friuli-Venezia-Giulia, Tuscany, the Apulia regions, and two Sicilian provinces). Moreover, the plan is to extend this work to four additional European countries thanks to a pilot study nested in an international Joint Action, namely, The Innovative Partnership for Action Against Cancer Joint Action and Joint Action PreventNCDs.

## 5. Conclusions

It is expected that further improvement in the interpretability of the BN model will be obtained while enriching it with more clinical and biological variables. Additional information on both treatments (e.g., drugs and doses) and lifestyle risk factors (i.e., body mass index, systolic pressure, cholesterol levels, and smoking) has already been collected for the CBC. In addition, in the ROSANNA project, a work package is analyzing preoperative circulating microRNAs to identify those associated with CVDs in patients treated in the NeoALTTO trial (NCT00553358) [29]. The circulating microRNA(s) identified in this dataset will be confirmed on independent case series collected in the prospective phase II NeoGENE trial (as the testing dataset), a multicenter, academic Italian study aimed at profiling HER2-positive tumor tissues before and after conventional trastuzumab-based therapy with the aim of studying tumor tissue and the plasma biomarkers of the response to treatment. Finally, the results will be validated using RWD extracted from a single-institution clinical cohort at the INT. The feasibility of the integration of microRNA results will also be assessed during the last year of the project (2025).

To conclude, in this paper, we propose an effective and innovative AI approach that makes it possible to compensate for data scarcity to overcome the missing-not-at-random and selection bias issues, also combining domain knowledge together with data gathered from different cohorts. This work could be a useful use case for all researchers dealing with RWD and rare events who aim at identifying and causally explaining the long-term outcomes in chronic patients.

## Figures and Tables

**Figure 1 cancers-16-03643-f001:**
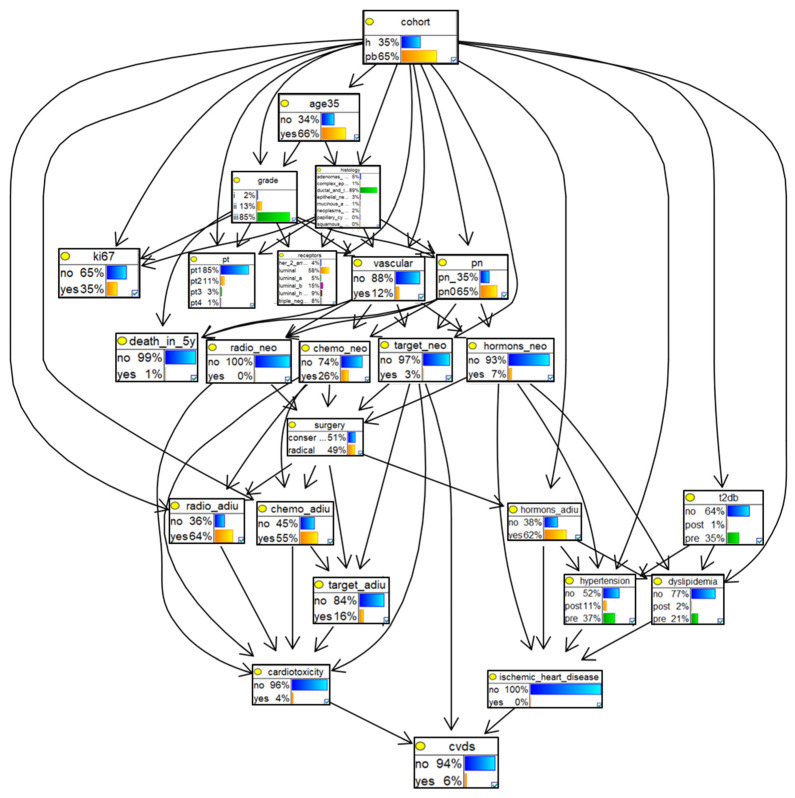
Final Bayesian network model structure. The figure illustrates the directed acyclic graph behind the model. Briefly, the cohort of origin is set as the root context node, followed by three subsequent lines that illustrate the cancer prognostic factors which affect both the 5-year survival and treatment decisions (neoadjuvant if pre-surgery, or adjuvant if post-surgery). Type 2 diabetes (t2db), dyslipidemia, and hypertension (bottom right in the figure) can already be present prior to the cancer diagnosis or can be induced by the cancer treatments. Finally, at the bottom of the model, the nodes can be seen describing cardiotoxicity induced by treatments and ischemic heart diseases, pointing to the target variable (i.e., cardiovascular diseases (CVDs)).

**Figure 2 cancers-16-03643-f002:**
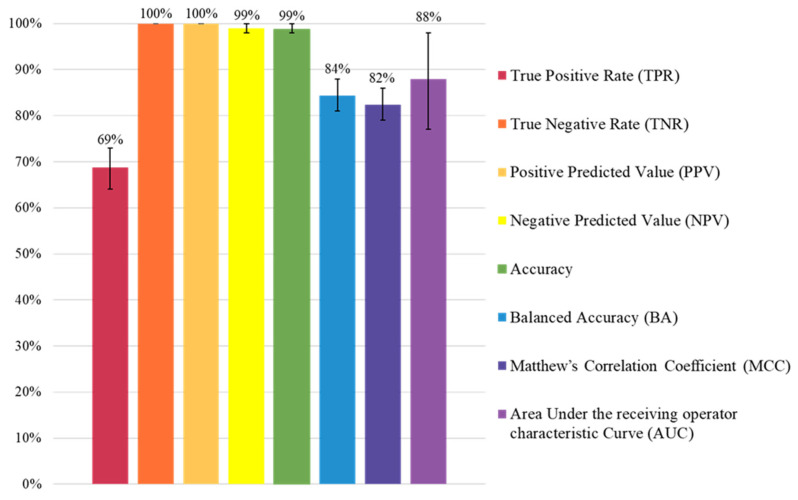
Bayesian network classification performances. Each classification metric is represented by a bar and the error bars are the 95% confidence intervals. Generally speaking, the model had good classification performances (AUC = 88% and accuracy = 99%). The model showed an almost perfect ability to correctly predict survivors not developing a CVD as not at risk (TNR, NPV), while identifying those developing a CVD as at risk is more complex due to the outcome rarity (TPR). When class imbalance is taken into consideration (BA and MCC), the performances decrease but remain very high (>80%).

**Figure 3 cancers-16-03643-f003:**
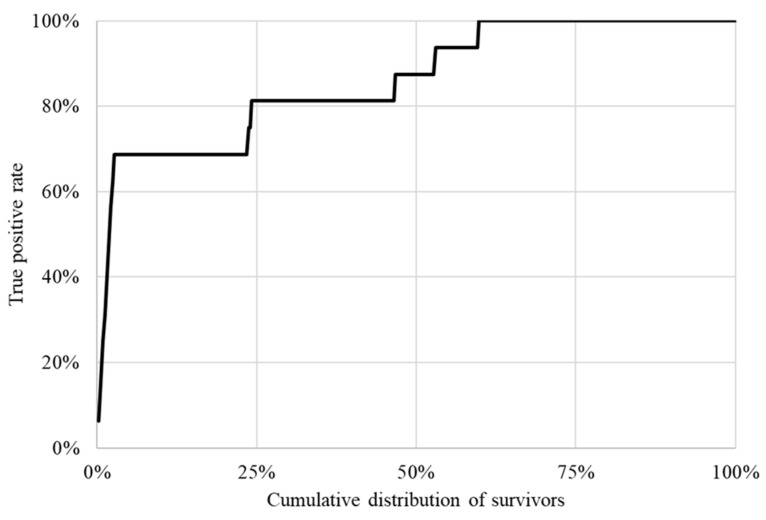
Lift chart obtained using Bayesian network model predictions. In this graph, survivors are sorted (*x*-axis), in decreasing order, according to their risk of developing a cardiovascular disease (CVD). While the cumulative distribution of the survivors increases, the corresponding True Positive Rate (i.e., the proportion of survivors in the validation set who actually developed a CVD within 5 years) is plotted on the *y*-axis. The sharp rise in the lift chart highlights the model’s good ability to identify survivors who should be prioritized for CVD follow-up.

**Figure 4 cancers-16-03643-f004:**
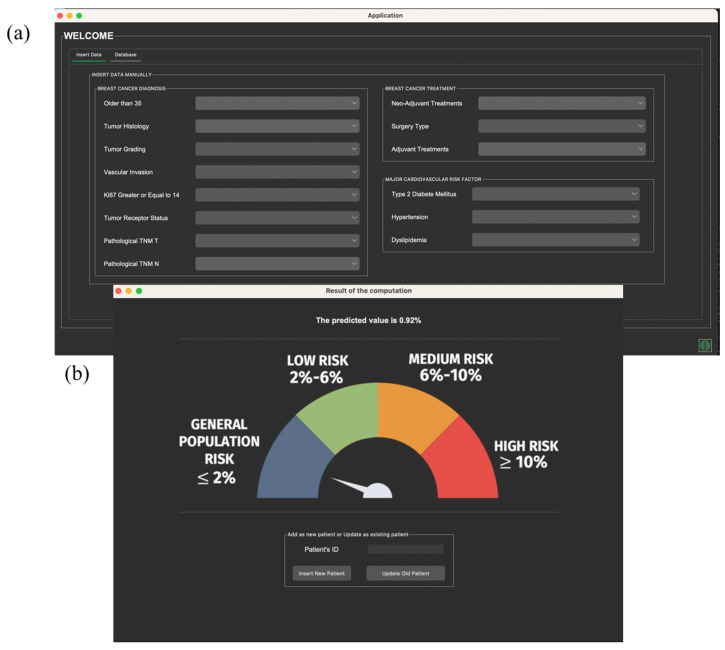
Concept draft of the interface (**a**) and the output (**b**) of the app-based tool. The figure illustrates the interface of the desktop-based app based on Bayesian network model probabilities. When the user inputs the patient’s information into the data entry window (**a**), the baseline model probability estimates for the 5-year cardiovascular disease risk are updated. Subsequently, in a new window (**b**), the user is provided with the individual predicted patient probability, together with an illustration that helps the user interpret the predicted value according to the risk classes.

**Table 1 cancers-16-03643-t001:** Variables used for the construction of the Bayesian network model.

Variable Type	Label in the Model	Description	Format	Values
Selection node				
	[cohort]	Cohort identification label	Discrete	Population-based cohort; clinic-based cohort
Cancer survival prognostic factors				
	[age35]	Patient aged 35 years or older at the date of breast cancer diagnosis	Dichotomous	Yes/No
	[histology]	Tumor morphology, according to ICD-O-3 ^1^ morphology codes	Discrete	Ductal and lobular neoplasm (ICD-O-3 M = 8500–8504, 8508, 8510, 8513, 8514, 8520–8523, 8530, 8540, 8541, 8543); epithelial neoplasms, NOS (ICD-O-3 M = 8010, 8015); adenocarcinomas (ICD-O-3 M = 8140, 8201, 8211, 8230); neoplasms, NOS (ICD-O-3 M = 8000, 8001, 8005); other histologies (ICD-O-3 M = 8050, 8070, 8575)
	[grade]	Tumor grading, according to ICD-O-3 codes	Discrete	Grade 1; Grade 2; Grade 3
	[vascular]	Tumor spread to the vascular system at the time of diagnosis	Dichotomous	Yes/No
	[ki67]	Ki67 index higher than 14%	Dichotomous	Yes/No
	[receptors]	Tumor receptor status	Discrete	Luminal; Luminal A; Luminal B; Luminal HER2; HER 2-enriched; triple-negative
	[pT]	Tumor size at the time of diagnosis (according to pathological stage)	Discrete	pT1 (<2 cm); pT2 (2–5 cm); pT3 (>5 cm); pt4 (spread to other organs)
	[pN]	Lymph nodes at the time of diagnosis (according to pathological stage)	Discrete	pN0 (no lymph node involvement); pN+ (lymph node involvement)
Cancer prognosis				
	[death_in_5y]	Death in the 5 years after cancer diagnosis	Dichotomous	Yes/No
Cancer treatments				
	[chemo_neo]	Neoadjuvant chemotherapy, i.e., the chemotherapy was administered within 6 months before the main surgical procedure	Dichotomous	Yes/No
	[radio_neo]	Neoadjuvant radiotherapy, i.e., the radiotherapy was administered within 6 months before the main surgical procedure	Dichotomous	Yes/No
	[target_neo]	Neoadjuvant target therapy, i.e., the targeted therapy was administered within 6 months before the main surgical procedure	Dichotomous	Yes/No
	[hormon_neo]	Neoadjuvant hormone therapy, i.e., hormone therapy was administered within 6 months before the main surgical procedure	Dichotomous	Yes/No
	[surgery]	Surgery type	Discrete	Conservative; radical (i.e., mastectomy)
	[chemo_adju]	Adjuvant chemotherapy, i.e., the chemotherapy was administered within 1 year after the main surgical procedure	Dichotomous	Yes/No
	[radio_adju]	Adjuvant radiotherapy, i.e., the radiotherapy was administered within 1 year after the main surgical procedure	Dichotomous	Yes/No
	[target_adju]	Adjuvant target therapy, i.e., the target therapy was administered within 1 year after the main surgical procedure	Dichotomous	Yes/No
	[hormons_adiu]	Adjuvant hormone therapy, i.e., the hormone therapy was administered within 1 year after the main surgical procedure	Dichotomous	Yes/No
Other cardiovascular risk factors				
	[dyslipidemia]	Diagnosis of dyslipidemia	Discrete	Pre (diagnosis of dyslipidemia within 1 year before the cancer diagnosis date); Post (diagnosis of dyslipidemia within 5 years after the cancer diagnosis date); No (no diagnosis of dyslipidemia)
	[hypertension]	Diagnosis of hypertension	Discrete	Pre (diagnosis of hypertension within 1 year before the cancer diagnosis date); Post (diagnosis of hypertension within 5 years after the cancer diagnosis date); No (no diagnosis of hypertension)
	[t2db]	Diagnosis of type 2 diabetes	Discrete	Pre (diagnosis of type 2 diabetes within 1 year before the cancer diagnosis date); Post (diagnosis of type 2 diabetes within 5 years after the cancer diagnosis date); No (no diagnosis of type 2 diabetes)
Target variables				
	[cardiotoxicity]	The patient showed any sign of cardiotoxicity (i.e., conduction disorders and arrhythmias or heart failure due to chemo-/radiotherapy) within 5 years after the cancer diagnosis	Dichotomous	Yes/No
	[ischemic_heart_disease]	Diagnosis of ischemic heart disease within 5 years after the cancer diagnosis	Dichotomous	Yes/No
	[cvds]	Diagnosis of any cardiovascular disease within 5 years after the cancer diagnosis	Dichotomous	Yes/No

^1^ ICD-O-3: International Classification of Diseases for Oncology, 3rd edition.

## Data Availability

Upon paper publication, a .net version of the model and the source code of the desktop application will be uploaded in a dedicated repository accessible via a web URL (https://github.com/madlabunimib/cardiovascular-diseases), together with a user manual, to allow readers to freely download them. Further requests for data sharing can be made to the corresponding author.

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
