# Peer review of "From Real-World Data to Causally Interpretable Models: A Bayesian Network to Predict Cardiovascular Diseases in Adolescents and Young Adults with Breast Cancer"

_cancers, 2024, doi:10.3390/cancers16213643_

Round 1

Reviewer 1 Report

Comments and Suggestions for Authors

Dear Authors,

In the manuscript entitled “From real-world data to causally-interpretable models: A Bayesian Network to predict cardiovascular diseases in Adolescents and Young Adults with Breast Cancer” the authors have tried to study the effects of AI-based model to identify AYA BC survivors at higher risk of developing a CVDs. My overall evaluation is negative. There are a number of major revision, formal and scientific aspects that should be addressed.

1.      In the introduction part, it is necessary to explain more about how to predict Cardiovascular Diseases by Bayesian Network. Mention points such as how to calculate the error percentage and validate the given results.

2.      Why the range of patients examined was between 2009-2014. Did you have a sample selection limit? Why didn't you consider a population for example between the years 2015-2023 to compare the two groups.

3.      Are the results obtained in the 2009-2014 patient population consistent with what is now available?

4.      The explanation of section "2.4. Model development and evaluation" is very confusing, it is necessary to make this section more understandable with a flowchart.

5.      Add more explanation below figure 1 to understand it.

6.      Add the necessary explanations under all the figures. Also, specify the comparison of groups with statistical methods, is it significant or not?

7.      Regarding the limitations of the method, it is necessary to add some content in the discussion section.

8.      It is necessary to add necessary explanations in the introduction and discussion about Breast Cancer and the possible risk of Cardiovascular Diseases. It is suggested to use the following reference to complete the article:

Moslemi, M., Vafaei, M., Khani, P. et al. The prevalence of ataxia telangiectasia mutated (ATM) variants in patients with breast cancer patients: a systematic review and meta-analysis. Cancer Cell Int 21, 474 (2021). https://doi.org/10.1186/s12935-021-02172-8

Author Response

In the manuscript entitled “From real-world data to causally-interpretable models: A Bayesian Network to predict cardiovascular diseases in Adolescents and Young Adults with Breast Cancer” the authors have tried to study the effects of AI-based model to identify AYA BC survivors at higher risk of developing a CVDs. My overall evaluation is negative. There are a number of major revision, formal and scientific aspects that should be addressed.

Comments 1: In the introduction part, it is necessary to explain more about how to predict Cardiovascular Diseases by Bayesian Network. Mention points such as how to calculate the error percentage and validate the given results.

Responses 1: We thank the reviewer for the suggestion. We introduced few sentences in the introduction to address this point.

Comments 2:  Why the range of patients examined was between 2009-2014. Did you have a sample selection limit? Why didn't you consider a population for example between the years 2015-2023 to compare the two groups.

Responses 2: We thank the reviewer for raising this point. In this project we took advantage of an already existing population-based cohort (PBC) we developed in Italy between in 2019 of about 70000 AYA cancer survivors. The cohort derives from population-based cancer registries (CRs). Each CR identified AYA cancer patients retrospectively. In this cohort, treatment for first primary cancer and all health events from diagnosis to death can be traced through linkage with available health databases, such as hospital discharge records, outpatient and pharmaceutical databases. In this cohort we selected only AYA with BC and we chose the time frame for diagnosis according to three major criteria:

  • The availability of the administrative data sources needed to define the needed variables (e.g., treatments): 2008 was the first year in which all these data sources were available, so, considering that we needed for all patients at least one year of observation before BC diagnosis to identify neoadjuvant treatment and CV risk factors, the first year of incidence available was 2009.
  • The minimum follow-up needed to observe the outcome: our work focuses on cardiovascular diseases within 5 years from diagnosis, the last follow-up available in the PBC was 2019, that’s why we had to stop the diagnosis period to 2014 in order to have for all patients at least 5 years of follow-up.
  • The consistency of the treatment within the diagnosis period: we reviewed the treatment guidelines and in Italy the BC treatment was consistent between 2009 and 2019. That’s why in the clinical cohort (in which we do not have any information on the follow-up) we considered the entire incidence period

We do not have any additional data for comparison but we will for sure validate the model in the future with a new data collection. We added a sentence about it in the paper.

Comments 3: Are the results obtained in the 2009-2014 patient population consistent with what is now available?

Responses 3:  We thank the reviewer for this important point. The results are consistent within the time frame 2009-2019 during which BC treatments did not change substantially in Italy, as explained in point 2. The major change in BC treatment in the more recent period was the introduction of immunotherapy that may impact the model substantially. We added a sentence in the discussion raising up this point.

Comments 4: The explanation of section "2.4. Model development and evaluation" is very confusing, it is necessary to make this section more understandable with a flowchart.

Responses 4:  We thank the reviewer for the suggestion, we added a flowchart in the supplementary materials, Figure S1.

Comments 5: Add more explanation below figure 1 to understand it.

Comments 6: Add the necessary explanations under all the figures. Also, specify the comparison of groups with statistical methods, is it significant or not?

Responses 5-6:  We thank the reviewer for pointing out the fact that explanations under the figures were missing, we added them. The objective of this work was to build an effective AI causal tool to predict the risk of cardiovascular diseases in AYA with breast cancer. We do not have a comparison group because it was not needed: we were not interested in making any formal comparison with other cancer survivors nor the general population.

Comments 7: Regarding the limitations of the method, it is necessary to add some content in the discussion section.

Responses 7: We thank the reviewer for the suggestion. We separated the limitations and the strengths from the general discussion and and further detailed the limits of the proposed methodology.

Comments 8: It is necessary to add necessary explanations in the introduction and discussion about Breast Cancer and the possible risk of Cardiovascular Diseases. It is suggested to use the following reference to complete the article:

Moslemi, M., Vafaei, M., Khani, P. et al. The prevalence of ataxia telangiectasia mutated (ATM) variants in patients with breast cancer patients: a systematic review and meta-analysis. Cancer Cell Int 21, 474 (2021). https://doi.org/10.1186/s12935-021-02172-8

Responses 8: We thank the reviewer for the suggestion, we revised the introduction and the discussion session to try to address the point. Moreover, we added two more references about it (including the one suggested by the reviewer) but we could not include any more reference because of the references limit to no more than 30 references.

Reviewer 2 Report

Comments and Suggestions for Authors

This is a well presented study demonstrating the development of a bayesian model to predict cardiotoxicity in breast cancer survivors - the study is very clinically relevant in tailoring follow up recommendations for this group - specifically those who need cardiooncology follow up - 

the complex methodology is clearly presented as are its limitations 

it would be helpful to discuss in the manuscript other studies in this area - neoallto is mentioned - are there others?

Comments on the Quality of English Language

line 284: not impossible should read not possible

Author Response

Comments: This is a well presented study demonstrating the development of a bayesian model to predict cardiotoxicity in breast cancer survivors - the study is very clinically relevant in tailoring follow up recommendations for this group - specifically those who need cardiooncology follow up - 

the complex methodology is clearly presented as are its limitations 

it would be helpful to discuss in the manuscript other studies in this area - neoallto is mentioned - are there others?

Responses: We thank the reviewer for the suggestion. We are not involved in other studies in this area and we are not aware about similar studies in AYA. However, we took the suggestion as an opportunity to further describe the task dedicated to the miRNA identification and validation.

Reviewer 3 Report

Comments and Suggestions for Authors

The authors proposed a Bayesian Network model, as well as an Artificial Intelligence model, to study and predict the 5-year risk of cardiovascular disease for young women breast cancer patients.

This is a well-written paper. I have some suggestions that can be used to improve the paper:

1. Is there any variable selection used in this analysis?

2. Methods for handling missing data need to be further discussed. You mentioned that the SEM algorithm is used, but why and how this method can handle MNAR needs to be discussed in detail. Also, since the outcome variable, cardiovascular disease, is completely missing in the CBC population, a sensitivity analysis that ignores the CBC data can be considered.

3. Death can be a competing risk of CVD, it is helpful to discuss this issue. 

4. One issue for this study is whether the developed model can be generalized to an independent population. One way to check is to fit the model using only data from parts of the CRs and use the remaining CRs as validation.

Author Response

The authors proposed a Bayesian Network model, as well as an Artificial Intelligence model, to study and predict the 5-year risk of cardiovascular disease for young women breast cancer patients.

This is a well-written paper. I have some suggestions that can be used to improve the paper:

Comments 1: Is there any variable selection used in this analysis?

Responses 1: We thank the reviewer for raising this point. The choice of the variables was done with the objective of ensuring the model causal explainability.  Bayesian Networks (BNs) offer several advantages when dealing with a low number of events (like CVDs in AYA) and multiple variables (as in our case). In such cases, traditional statistical methods may struggle due to data sparsity, but Bayesian Networks can still perform well due to their probabilistic framework and ability to handle uncertainty, incorporating prior knowledge in the form of prior probabilities. We added a sentence about the variables’ selection in Section 2.3.

Comments 2: Methods for handling missing data need to be further discussed. You mentioned that the SEM algorithm is used, but why and how this method can handle MNAR needs to be discussed in detail. Also, since the outcome variable, cardiovascular disease, is completely missing in the CBC population, a sensitivity analysis that ignores the CBC data can be considered.

Responses 2: We thank the reviewer for the suggestion. Section 5.2 of “Nir Friedman. 1998. The Bayesian structural EM algorithm. In Proceedings of the Fourteenth conference on Uncertainty in artificial intelligence (UAI'98). Morgan Kaufmann Publishers Inc., San Francisco, CA, USA, 129–138” discuss how to use the SEM algorithm with MNAR. We revised the methods accordingly.

Comments 3: Death can be a competing risk of CVD, it is helpful to discuss this issue. 

Responses 3: We thank the reviewer for pointing out this important point. Death is for sure a competing event in this analysis, that’s why, to control for it we added the node “Death in 5 years” in the model. We added a sentence about in the results.

Comments 4: One issue for this study is whether the developed model can be generalized to an independent population. One way to check is to fit the model using only data from parts of the CRs and use the remaining CRs as validation.

Responses 4: We thank the reviewer for the comment. This is what we did, indeed. We split the overall combined data set (PBC+CBC) into two subsets: 339 patients from the CBC + 623 patients from the PBC (approx. 70% of the overall sample) contributed to the training set, while the remaining 413 patients from the PBC (approx. 30% of the overall sample) contributed to the test set that we used for validation. We modified the name of the test set into validation set and made the process clearer adding a flow-chart to the supplementary materials.

Round 2

Reviewer 1 Report

Comments and Suggestions for Authors

The authors' answers to the questions were convincing. I think the article is acceptable.

Reviewer 3 Report

Comments and Suggestions for Authors

The authors have addressed all my comments and I do not have further questions.